# The Combined Effect of Low-dose Atropine with Orthokeratology in Pediatric Myopia Control: Review of the Current Treatment Status for Myopia

**DOI:** 10.3390/jcm9082371

**Published:** 2020-07-24

**Authors:** José-María Sánchez-González, Concepción De-Hita-Cantalejo, María-José Baustita-Llamas, María Carmen Sánchez-González, Raúl Capote-Puente

**Affiliations:** 1Department of Physics of Condensed Matter, Optics Area, University of Seville, 41012 Seville, Spain; mhita@us.es (C.D.-H.-C.); mbautista5@us.es (M.-J.B.-L.); msanchez77@us.es (M.C.S.-G.); rcapote1@us.es (R.C.-P.); 2Department of Ophthalmology & Optometry, Tecnolaser Clinic Vision, 41018 Seville, Spain

**Keywords:** atropine, orthokeratology, myopia control, refractive errors, peripheral refraction, myopia progression

## Abstract

Pediatric myopia has become a major international public health concern. The prevalence of myopia has undergone a significant increase worldwide. The purpose of this review of the current literature was to evaluate the peer-reviewed scientific literature on the efficacy and safety of low-dose atropine treatment combined with overnight orthokeratology for myopia control. A search was conducted in Pubmed and Web of Science with the following search strategy: (atropine OR low-dose atropine OR 0.01% atropine) AND (orthokeratology OR ortho-k) AND (myopia control OR myopia progression). All included studies improved myopia control by the synergistic effect of orthokeratology with low-dose atropine, compared with orthokeratology treatment alone. All studies included a short or medium follow-up period; therefore longer-term studies are necessary to validate these results.

## 1. Introduction

Pediatric myopia has become a major international public health concern [1]. The prevalence of myopia has increased significantly worldwide [2]. Accordingly, prevention and visual care need to increase in order to control the risk of developing eye diseases such as retinal detachment, glaucoma, cataracts, and myopic macular degeneration, which may lead to irreversible vision loss [3].

Juvenile-onset myopia is the most common variety [4], with its peak incidence during elementary school and progression through the teenage years [5]. Thus, it is crucial to make a myopia onset prediction between the ages of 8 and 9 years old [6,7]. Myopia has been described as the sixth most common vision loss cause [8], after ocular pathologies such as age-related macular degeneration, glaucoma, cataracts, and diabetic retinopathy. For this reason, treatments to slow the progression of myopia have become an essential area of research. Among the myopia control options are progressive ophthalmic lenses, different dose topical atropine, overnight orthokeratology lenses, and soft multifocal contact lenses [9,10]. The primary goal of these treatments is to reduce myopia progression in order to decrease the high myopia blindness rate. Recently, synergistic combined treatments have been investigated as new research lines. The success rates of both orthokeratology and low-dose atropine treatments are less than 100%; therefore, because the mechanisms of these two treatments are almost certainly different, it is possible that the efficacy of these treatments in combination might be greater than that of either one alone. To the best of our knowledge, our review is the first to evaluate the peer-reviewed scientific literature on the efficacy and safety of low-dose atropine treatment combined with overnight orthokeratology for myopia control.

## 2. Mechanism of Action

### 2.1. Atropine

Recent studies provide clinical and experimental evidence that inflammation plays a crucial role in the development of myopia [11]. The relationship between myopia and axial length has been widely studied [12]. Homeostatic eyeball growth control occurs through chemical signals, which shape and mold the sclera and choroid. Vision is the main input and causes a chemical signaling cascade from the retina to the retinal pigment epithelium to the choroid to the sclera, which regulates ocular growth [13,14,15]. Structurally, the sclera is made up of collagen, elastic fibers and fibroblasts. Fibroblasts synthesize type I collagen, the main component of the sclera [6]. A decrease in the amount of collagen causes a thinning and lengthening of the sclera. There are studies that reveal the presence of muscarinic receptors in the sclera. Atropine is a non-selective antagonist of the muscarinic acetylcholine receptor (mAChR). From the pharmacological point of view, the receptor blockade stops the proliferation of scleral fibroblasts and the consequent axial lengthening of the eye [16].

The effects of atropine on the choroid and retinal pigment epithelium have also been reported [17,18]. Myopia associates with chronic inflammation of the eye and could be controlled by atropine or with biochemical changes at the sclera level that interfere with ocular development [11,19]. Several articles agree that certain molecular and biochemical events take place in the sclera during the remodeling process, causing structural and biomechanical changes in the tissue and significant thinning of scleral tissue [15,20].

Atropine is used topically, administered as drops in different concentrations for myopia control, as follows: low dose—0.01%, moderate dose—between 0.01% and 0.5%, and high dose—between 0.5% and 1.0% [21]. The 0.01% atropine formulation is marketed under the name Myopine ™ [22]. This product is currently available in Singapore, Malaysia and has been licensed in Asia and in 15 countries in Europe.

### 2.2. Orthokeratology

The overnight orthokeratology lenses treatment (OK) has an inverse geometry design, in which the base curve or treatment area is flatter than the corneal apex. This design allows the corneal central zone to flatten due to the positive pressure exerted on the corneal apex, while the medial corneal periphery remains more pronounced. In this way, the normally prolate corneal profile becomes a spherical and even oblate shape. Thus, peripheral hypermetropy blurring is achieved [23]. The OK standard design is three curves from the center to the periphery. OK results in a temporary flattening of the central cornea, which corrects the central myopic refraction but also shifts the peripheral refractive status from hyperopic defocus to relative myopic defocus. Peripheral refractions achieved in children wearing OK have been verified, as well as a reduction in central myopic refraction [24]. In order to improve lens centration, the lens design can be modified by introducing an alignment curve between the reverse curve and the peripheral curve. This alignment curve may also be divided into two curves to form a five-zone lens [25]. Regarding the material, higher oxygen transmissibility improves the effectiveness of OK [26]. It increases epithelial cell metabolism by receiving more oxygen, and faster epithelial thinning occurs. However, this effect disappears after two weeks of treatment, making it unlikely that the lens material affects changes in the corneal shape [23]. Most OK designs are spherical. In recent years, the toric OK design has emerged. These lenses achieve slower myopia progression among users with moderate to high astigmatism [27] and also improves the OK centration during treatment [28].

## 3. Low-Dose Atropine Efficacy

Table 1 details data from three articles that studied the atropine efficacy in decreasing axial length in a treatment group vs. a control group. Yam et al. [29] evaluate the efficacy and safety of low-concentration atropine eye drops at 0.05%, 0.025%, and 0.01% compared with placebo over a 1-year period. Of the three concentrations used, 0.05% atropine was the most effective in controlling SE progression and AL elongation over a period of 1 year. However, the cessation of atropine doses of 0.1% and 0.5% were related to the greater myopia rebound and an increase in the axial length values. There is evidence of the ability of atropine to slow down axial lengthening and childhood myopia progression. Higher concentrations have been shown to be more effective but also have a greater rebound effect after stopping treatment [30]. Furthermore, in these studies, the age of children varies as well as the myopia and astigmatism values [29,31,32]. However, after the review, we can affirm that in terms of refraction orthokeratology and in terms of axial length, atropine is the most effective in all populations.

Efficacy in terms of axial length changes is presented in Table 1. Almost all studies studied Asian populations and therefore evidence on European and other ethnicities is limited [33,34] although there appears to be a difference in efficacy between white and Asian children [34] that may be justified by the greater progression of myopia in Asian children and not by the efficacy of the drug. However, dosing guidelines and populations have not been well established in some of the studies consulted; therefore, further investigation is required. Additionally, the treatment should be adapted to each patient. Investigation of the long-term efficacy and safety through randomized clinical trials might be necessary to confirm these results.

## 4. Overnight Orthokeratology Efficacy

OK control myopia efficacy has been discussed in this last decade by several articles that generally showed a decrease in axial length in a treatment group versus a control group [37,38,39,40] (Table 2). Recently, a Chinese study reported the control of myopia progression, particularly in cases with high myopia and axial length elongation. After one year, the average axial elongation was 0.27 ± 0.17 mm in the ortho-k lens group and 0.38 ± 0.13 mm in the control group, so the axial elongation in the ortho-k group was 57.1% slower than that in the control group [41]. Sun et al. [42] published a meta-analysis in which they evaluated the effectiveness of OK and assessed its effects in slowing myopia progression. Their results found a 0.14 mm/year deceleration in axial length in the OK group compared with the control group with a decrease in the progression of myopic error of approximately 45%. Zhou et al. [43], Lee et al. [44], and Hiraoka et al. [45] analyzed longer periods of 6, 7, and 10 years, respectively. They observed the changes generated by OK in young patients and concluded that the refractive myopic error decreased significantly. Hiraoka et al. [45] found a rebound phenomenon after 10 years of OK treatment.

Similarly, a study with a Caucasian population [46] compared OK axial length growth against a control group for seven years, finding a tendency for the axial lengthening to decrease in the order of 33% in the OK group in comparison with the control group. In the last year [47,48], several literature reviews have shown that OK is effective for myopia control. Moreover, the overnight wear mode could create comfort for those who do not like wearing glasses or carrying contact lenses during the day. This also provides freedom for participation in sports activities. A new study [49] reviewed different works on the effects of OK, reporting a better quality of life and satisfaction with vision by patients who completed a visual scale. Findings have shown orthokeratology to be effective in slowing axial elongation. It is a more well-accepted treatment than the use of single vision glasses.

## 5. Efficacy of Low-Dose Atropine Combined with Orthokeratology

Since OK and atropine both slow down the progression of myopia, there could be a synergic therapeutic effect when applied in combination. A search was conducted in PubMed and Web of Science with the following search strategy: (atropine OR low-dose atropine OR 0.01% atropine) AND (orthokeratology OR ortho-k) AND (myopia control OR myopia progression). This search achieved forty-two results. Thirty-nine of them were studies conducted in the last decade and twenty-nine were conducted in the last three years. After analyzing the abstracts and titles, only four articles [51,52,53,54] were clinical studies on the AT and OK combination. A fifth Russian article [55] was found and was not included for language reasons. Despite there being a limited number of studies, these four studies have been published in the last twenty-four months, showing the current scientific interest in using combination therapies to slow myopia progression.

Regarding reviewed articles, after three months of OK treatment stabilization, Kinoshita et al. [51] randomly allocated children to an AT + OK group or an OK alone group. Low-dose atropine was applied once daily at night. In the study, the corneal endothelial cell density (CECD), intraocular pressure (IOP), and uncorrected distance and near-visual acuity (UDVA and UNVA) were controlled. The authors did not find significant differences in CECD or IOP between individuals treated with AT and OK vs. those in the OK group alone. They also reported a non-significant correlation between AXL and age but a strong correlation between AXL and spherical equivalent refraction (SER) (r = 0.805, *p* < 0.01). A few months later, Wan et al. [52] described four sub-section studies. They split the sample into patients under and above six diopters (D) and also allocated individuals to treatment with 0.125% AT with OK and 0.025% AT with OK. All groups were compared with a control OK-alone group. Children were given one AT drop daily, and OK lenses were worn before sleep. No significant differences in accommodation, photopic pupil diameter, or mesopic pupil diameter were found at baseline. In addition to the two studies discussed above, Chen et al. [54] divided their study into two phases. In the first one, participants underwent OK treatment only and, after twelve months, 0.01% AT was added. They concluded that patients with higher myopia changes could be benefit from a low dose of AT. The most recent study was published by Tan et al. [53] This only provided premilitary results and the follow-up period was only one month; however, they found statistically significant differences in AXL between a group given AT with OK against a group given OK alone.

The methodological characteristics of the four studies and their main results are presented in summary form in Table 3. Patients were aged between 8.3 and 10.9 years old. The mean patient age was 10 years old. Before treatment, the range of refraction was from −2.65 D to −6.75 D. The mean pre-treatment refraction was −4.34 D. The follow-up period ranged from only one month to twenty-four months, and the mean follow-up period was 19 months. For the treatment group, all studies used low-dose atropine (0.01%, 0.025%, or 0.125% concentration) with simultaneous OK treatment, while the control group received OK alone in all cases. The number of children included in the treatment group ranged from 20 to 60 with a mean of 31. The number of patients in the control group ranged from 20 to 35 with a mean of 25.5 children. The mean refraction increase in the treatment group was 0.38 D, while the control group who only received OK achieved near to double this increase with 0.62 D. In terms of uncorrected distance visual acuity (UDVA), the treatment and control groups obtained excellent results, and a non-significant difference was found for this issue. Finally, for the axial length, as the most important concern in myopia control assessment, it was revealed that the group that received low-dose AT with OK only had an AXL increase of 0.37 mm, while an increase of 0.41 mm was reported for the OK-alone group. Statistically significant differences (*p* < 0.05) were found in all clinical studies [51,52,53,54].

All studies showed improved myopia control by the synergistic effect of OK with low-dose AT concentration, compared with OK therapy alone. The potential mechanism combination was based on pupil mydriasis by increased retinal illumination due to AT, which would induce a shorter myopic change in the peripheral retina and upgrade the effectiveness of OK treatment [52]. Previous authors proved that the light cycle is related to AXL growth and maturation [56]. The results obtained by all studies for combined treatment demonstrated better efficacy in patients with low myopia. Although Tan et al. [53] found clinically insignificant AXL changes, finding statistically significant differences in the AXL elongation in just one month can give us an idea of the effects that this combination could have in the long term. Recently, Zhao et al. [57] studied the one-month change in subfoveal choroidal thickness of myopic children treated with 0.01 atropine and OK and they found that the combination of atropine and OK revealed a greater increased in subfoveal choroidal thickness than monotherapy of atropine alone, which indicates a better myopia control treatment option. All studies included a short or medium follow-up period, so longer-term studies are needed to validate these results.

## 6. Safety and Complications

### 6.1. Atropine

Treatment with atropine and treatment with orthokeratology have been associated with complications. The main reported atropine complications in myopia control are photophobia and near vision blur, as a result of increased pupil size and decreased accommodative amplitude. This effect is directly proportional to the atropine concentration used. In a 0.01% atropine concentration study, Wu et al. [58] found that only 7% of the participants presented photophobia. Pupil dilation, accommodation amplitude (0.05% eyes (13.24 ± 2.72 D)), and near vision loss [30] were minimal with 0.01% atropine, but they were significant compared to patients who did not have this treatment [35]. This impact returned to untreated levels two months after treatment interruption [59]. No severe side effects with long-term atropine use have been found in addition to those described. Allergic conjunctivitis and contact lenses management difficulties have very low incidence rates [51,60]. An increase in pupil size with atropine could cause increased aberrations and a decrease in visual quality; however, children with larger pupil sizes would receive a greater proportion of peripheral myopic defocus associated with OK [52,54,61]. As for the rebound effect when the treatment is discontinued [30,62], Lixia et al. [63] found −0.72 ± 0.75 D after 36 months.

The side effects of the use of a low dose of atropine (0.01%) that have been observed so far occur in a small percentage of patients and do not generally require withdrawal of treatment.

### 6.2. Orthokeratology

Regarding OK, the change in corneal geometry causes an increase in ocular aberrations [64,65]. Chen et al. [61] described a statistically significant rise in spherical aberrations (0.05 ± 0.6 to 0.19 ± 0.13 µm) and coma aberrations (0.12 ± 0.19 to 0.36 ± 0.23 µm). Other authors found similar results and conclusions [49,66,67,68,69]. Some authors related this increase in aberrations with the decrease in the corneal thickness [70] with contact lens use of approximately 20 µm [61], although Guo et al. [71] did not consider these significant changes after seven years of follow-up, coinciding with other studies that did not find changes in the endothelial cell density [72,73].

The increase in aberrations described in OK users does not seem to have a major impact on subjective vision, suggesting that neural adaptation is sufficient to overcome optical quality degradation [74], [66] and, furthermore, this change in aberrations, particularly in spherical aberrations, induces a greater accommodative response that would help to compensate for them [69]. This adaptation is faster in children than in the adult population [74].

The most severe OK complication is microbial keratitis, although it does not present a pattern of clear onset; rather, it occurs sporadically, related to poor contact lense hygiene [75]. Acanthamoeba and Gram-negative bacilli, especially *Pseudomonas aeruginosa*, are the most common pathogens [76], although most cases are not resolved satisfactorily. Complications include corneal opacity in all eyes, glaucoma in one eye, and cataract in one eye.

Bullimore et al. [48] recently concluded that microbial keratitis could occur in Asian children who wear contact lenses, and it was not possible to estimate the frequency or incidence of these events in other countries because the majority of papers on the matter refer to these populations. Moreover, no microbial keratitis incidence differences versus soft contact lenses have been reported [77,78,79]. Other complications present in OK users are epithelial iron deposit, white lesions, and fibrillary lines, which are associated with the duration of the treatment [80].

Moreover, corneal staining, a steep fitting lens, and tear film stability are related to nightly use [80]. OK is highly likely to also have a rebound effect after discontinuation [47,81]; however, this has not yet been investigated in combination with AT. The combination of orthokeratology and atropine in low concentrations does not have a greater impact on complications than those presented separately [53]. Studies indicate that orthokeratology lenses are safe and have no adverse physiological effects on the cornea and vision. In case of adverse effects, interruption of the use of the lenses can return the cornea to its original shape.

## 7. Future Research

### 7.1. When to Interrupt Treatment for Myopia Control?

The increased incidence of myopia has led to advances in the search for lens and instrument design for myopia control. This is expanding the possibility of correcting more cases of astigmatism and myopia [82]. Regarding treatment efficacy, the actual control mechanism and knowledge of the rebound effect are possible future lines of research. Cho and Cheung [81] compared the changes in axial length in different subjects after OK discontinuation and subsequent resumption compared to patients who continued to wear OK. They reported doubts about the benefits over time and questioned whether discontinuation is associated with myopic acceleration following OK therapy. However, new research is needed related to the timing or line of treatment discontinuation on myopia control.

### 7.2. New Spectacles Design

Research interest in the past decade has focused on the use of different treatments to potentially slow the progression of myopia. The use of glasses to delay the progression of myopia is another form of management, but despite its high tolerance and easy adaptation, the results currently obtained are not statistically significant [83]. A three-year clinical trial reported the effects on myopia progression deceleration with bifocal lens use with a base-in prism and its efficacy in children with accommodative lag depending on the phoria state function [84]. Animal studies have provided solid evidence that imposed myopic defocus (MD) inhibits eye growth. A spectacle lens based on the MD mechanism for myopia control has been designed, and it is named the Defocus Incorporated Multiple Segments (DIMS) spectacle lens. It imposes MD on both the central and peripheral retinas. Recently, Lam et al. [85] used lenses (DIMS) based on the constant myopic blur principle for myopia control, demonstrating a 52% reduction in myopic evolution and a 62% reduction in axial length in the DIMS lens group.

### 7.3. Daily Soft Contact Lenses

In children and adolescents, to control myopia, the use of daily contact lenses is recommended for their safety [86] and easy handling. A recent study [87] involved a clinical trial of 74 children—41 in the daily soft lens group and 33 in the single-vision group. The results showed slower progression in the double-focus lens group (axial length change of 0.28 mm) compared with the control group (axial length change of 0.44 mm). In addition, Chamberlain et al. [88] showed, after a three-year follow-up period, similar efficacy in slowing down myopia progression with daily use of soft contact lenses.

### 7.4. Outdoors Activities and Sunlight Exposition

Increased outdoor time has been shown to have a positive influence on myopia progression, although it is not entirely effective in eyes that are already nearsighted [89].

The mechanism by which outdoor activity could help to prevent the development of myopia remains unclear. A meta-analysis showed that a greater amount of time spent outdoors is associated with a lower myopia increase. They reported a decrease of 2% for each additional hour spent outdoors per week [90]. An increase in daily outdoor activity with strong sunlight exposure may not necessarily be a prevention factor in myopia control.

## 8. Conclusions

The available scientific literature demonstrates that combined treatment of low-dose atropine with simultaneous overnight orthokeratology is effective and safe for myopia control. However, since efficacy has a synergistic effect, complications and management are also additive when both treatments are used; therefore, more long-term studies with a multidisciplinary analysis are needed to investigate myopia control behavior in populations other than just Asian populations. Currently, available evidence demonstrates that this is a promising research line that needs longer-term randomized clinical trials to confirm the outcomes shown.

## Figures and Tables

**Table 1 jcm-09-02371-t001:** Study characteristics and results summary of low-dose atropine studies.

Author (Year)	Patients	Follow-up (Months)	Treatment		Control	
			Atropine	Δ AXL (mm)	Group	Δ AXL (mm)
Fu et al. [35] 2020	142	12	0.01%	0.37 ± 0.22 *	SVG	0.46 ± 0.35 *
Azuara-Blanco et al. [33] 2019	289	24	0.01%	NR	Placebo	NR
Yam et al. [29] 2019	438	12	0.01%	0.36 ± 0.29 *	Placebo	0.41 ± 0.22 *

AXL: Axial length; mm: millimeters; SVG: single vision glasses; NR: not reported; OK: orthokeratology; AT: atropine * statistically significant difference *p* < 0.05. Chia et al. [30,36] reported intermittent atropine treatments.

**Table 2 jcm-09-02371-t002:** Study characteristics and results summary of orthokeratology myopia control.

Author (Year)	Patients	Follow-up (Months)	Treatment Group (OK)		Control		
			Δ SE (D)	Δ AXL (mm)	Group	Δ SE (D)	Δ AXL (mm)
Santodomingo-Rubido et al. [46] 2017	30	84	0.29 ± 0.10 *	0.91 ± 0.27 *	SVG	−5.00 ± 0.43 *	1.35 ± 0.27 *
Lee et al. [44] 2017	102	24	0.17 ± 0.02 *	NR	SVG	−0.52 ± 0.03 *	NR
Hiraoka et al. [45] 2018	92	120	−1.,26 ± 0.98 *	NR	SCL	−1.79 ± 1.24 *	NR
Lyu et al. [50] 2020	247	24	NR	0.58 ± 0.35	AT	NR	0.36 ± 0.30 *

OK: Orthokeratology; SE: Spherical equivalent; AXL: Axial length; mm: millimeters; SVG: single vision glasses; SCL: single contact lens; NR: Not reported. * Statistically significant difference *p* < 0.05.

**Table 3 jcm-09-02371-t003:** Study characteristics and results summary of the four articles included in the low-dose and orthokeratology section.

Author (Year)	Age (Years)	Rx (D)	Follow-up (Months)	Treatment Group					Control Group		
				Combination (n)	∆ Rx (D)	Post VA (LogMAR)	∆ AXL (mm)	∆ AXL (mm)	Therapy Alone (n)	∆ Rx (D)	Post VA (LogMAR)
Kinoshita et al. [51] 2018	10.6	−2.88	12	0.01% AT + OK (20)	NR	NR	0.09 ± 0.12 *	0.19 ± 0.15 *	OK (20)	NR	NR
Wan et al. [52] 2018	10.4	−4.25	24	0.125% AT + OK (20)	↑ 0.50 D	0.01 ± 0.01	0.55 ± 0.12 *	0.58 ± 0.09 *	OK (26)	↑ 0.55 D	0.01 ± 0.01
Wan et al. [52] 2018	10.3	−4.58	24	0.025% AT + OK (20)	↑ 0.30 D	0.01 ± 0.01	0.65 ± 0.18 *	0.83 ± 0.16 *	OK (20)	↑ 0.83 D	0.01 ± 0.01
Wan et al. [52] 2018	10.9	−6.75	24	0.125% AT + OK (24)	↑ 0.25 D	0.01 ± 0.01	0.57 ± 0.17 *	0.64 ± 0.14 *	OK (29)	↑ 0.45 D	0.01 ± 0.01
Wan et al. [52] 2018	10.8	−6.48	24	0.025% AT + OK (20)	↑ 0.49 D	0.01 ± 0.00	0.58 ± 0.08 *	0.40 ± 0.15 *	OK (20)	↑ 0.65 D	0.01 ± 0.00
Chen et al. [54] 2018	8.3	−2.65	24	0.01% AT + OK (60)	NR	NR	0.14 ± 0.14 *	0.25 ± 0.08 *	OK (29)	NR	NR
Tan et al. [53] 2019	9.0	−2.79	1	0.01% AT + OK (33)	NR	−0.04 ± 0.07	−0.05 ± 0.05 *	−0.02 ± 0.03 *	OK (35)	NR	−0.03 ± 0.07

Rx: Refraction; D: Diopters; VA: Visual acuity; LogMAR: Logarithm minimal angle resolution; AXL: Axial length; mm: millimeters; AT: Atropine; OK: Orthokeratology; NR: Not reported. * Statistically significant difference *p* < 0.05; ↑ = increased. The work by Wan et al. [52] 2018 was split into four because they studied two populations, under 6 diopters and above 6 diopters, and two atropine concentrations, 0.125% and 0.025%.

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
