# Peer review of "The Combined Effect of Low-dose Atropine with Orthokeratology in Pediatric Myopia Control: Review of the Current Treatment Status for Myopia"

_jcm, 2020, doi:10.3390/jcm9082371_

Round 1
Reviewer 1 Report
This paper addresses a significant issue, but it has a number of limitations that need to be addressed.
The first is that the English is sometimes rather strange, and sometimes misleading. I recommend that the authors get assistance from a native English speaker or a professional service to improve the wording.
There is also a problem with the search strategy used, because it seems to equate “low dose atropine” with 0.01%. However, the recent work from the LAMP study suggests that 0.05% atropine is likely to emerge as the dose that best balances efficacy and side-effects. As far as I am aware, there are no studies using 0.05% atropine in combination with orthokeratology, which therefore leaves a major gap in the literature. The authors justify the emphasis on 0.01% atropine, by referring to a study by Tian et al showing toxicity to corneal cells about 0.03%. However, this comparison has very limited validity, because topical application of a drop of 0.05% atropine is unlikely to lead to effective concentrations of that level, except perhaps extremely transiently. Since, with very much higher doses of atropine, there have been no reports of corneal changes, the logic used is not at all compelling.
The results in the paper consist of three tables. The first concerns studies using 0.01% atropine. This area has been extensively, and in my opinion better, reviewed in several publications. I have already raised the atropine dose issue, and it does not strike me as useful to include studies that have not yet reported results.
The second table covers studies on OK, and again the subject has been extensively review already.
Table 3 covers studies that have used a combination of 0.01% atropine and OK. From the presentation, it is not easy to see that the synergistic effect is better than that of OK alone. It would be better to present the OK alone effects together with the combination.
If there is, as I have suggested, a shift to 0.05% atropine as the standard dose, it will be necessary to try the new combinations, preferably in comparison to atropine or OK alone.
The scientific analysis in the paper is also quite low. I will give only a few examples.
Lines 29-31: The issues around race and ethnicity are complex, but it does not appear that there are fundamental differences based on genetics. The presentation does not address any of these issues.
Lines 31-32: Hong Kong is generally not regarded as a country, but as an SAR of China. Opinions are more mixed about the status of Taiwan, but few countries regard Taiwan as an independent country, whatever they think about rapid union with mainland China.
Lines 34-36: The discussion of risk factors is confused. Most, but not all people would agree about nearwork, but there is more controversy about digital devices. There is a complex overlap of time outdoors, daily light and UV exposures. Time outdoors in brighter light and UV exposures run pretty much in parallel, but it appears that it is exposure to light that is the important factor. Higher consumption of saturated fats increases axial length and probably overall body stature, but does not change refraction. And what do the authors mean by “the genetic factor.” This review is not useful, since it does not clarify issues, but is more likely to spread confusion. It is not really required for the topic of the paper.
Author Response
We would like to thank reviewers for their insightful comments on the paper, as these comments led us to improve it. We took into consideration all reviewers comments and addressed them in the paper. Detailed responses to reviewers are given below.

Reviewer 2 Report
Please see attached reviews.

Author Response

(The authors gave the same response as above.)

Round 2
-
Reviewer 1 Report
The authors have made some major changes to this paper, but it still has severe limitations.
In relation to Table 1, there is no point in listing a study that has not yet presented results. There are also three studies from China reported, with unspecified time lengths. This seems extraordinary. Have the authors had the papers translated, or have they relied on an abstract? Of these studies, 2 report big effects and one reports almost no effect. If we exclude these studies, there are only 2 that can be compared. These report only modest effects (13% and 20% reduction). The authors need to deal in more detail with the study by Yam et al, which reported on 0.05% atropine. This concentration had a much greater effect (around 67% reduction), and in my opinion seems likely to become the standard dose used. But clearly, more work on the best dose is needed.
In relation to OK, there seems to be considerable variation in the efficacy. The paper by Lyu et al seems to report a negative effect. Can the authors check to make sure they have got all values correct? What have meta-analyses of OK results reported.
In relation to combinations, the effects seem to be all over the place, and I do not think that the simple conclusion that combinations work is justified.
The background to this paper is not strong.
The authors cited a paper by Zadnik et al on increases in the prevalence of myopia world-wide, but this is a paper on prediction of myopia onset from one set of data. The next paper cited is that of Holden et al., which is more relevant, but it is a set of projections, based on a predictive model that I believe is highly flawed.
I do not understand what the authors are trying to say in lines 31-32, and why it is crucial to make predictions at the age of 8 or 9.
The authors then state that “the slow progress of myopia treatments makes this an essential area for research.” IF the authors mean the limited progress in developing preventive myopia treatments, then I think this gives a false picture of the literature. With OK, 0.05% atropine, MiSight contact lenses and DIMS spectacles (with two other commercial versions about to be released), I think the reverse is true.
The authors then claim that the mechanisms underlying the effects are almost certainly different. I disagree. There are likely to be final common pathways, as is seen for the apparently diverse effects of FDM, LIM and atropine at the scleral level. For optical interventions to affect axial elongation, they need to results in biochemical changes in cellular pathways, and these may be common to all interventions.
The section on the mechanism of action of atropine is confused. The leading possibility is that atropine produces changes in the IPL, which results in suppression of the increases in EgR1 which are seen when axial elongation increases. Everything flow from there. But it is not clear that the receptors are muscarinic (see reports by Stell and colleagues and Ashby and colleagues).Nickla et al have reported direct effects of atropine on scleral fibroblasts.
I think the literature currently regards 0.01% and 0.05% atropine as low doses.
The authors state that previous studies showed that 0.01% atropine did not significantly decelerate axial elongation, but Table 1 reports on two studies that did show significant differences. Something does not make sense.
The claimed difference between Asian and White children is there when absolute changes are considered. But when percentage changes are considered, the effects are the same. This is simply due to the greater progression seen in Asian children, and is not a fundamental difference in drug efficacy.
In summary, this paper is still confused and incomplete, although it has improved.
Author Response#RV0: The authors have made some major changes to this paper, but it still has severe limitations.
#AU0: We would like to thank reviewers for their insightful comments on the paper, as these comments led us to improve it. We took into consideration all reviewers comments and addressed them in the paper. Detailed responses to reviewers are given below.
#RV1: In relation to Table 1, there is no point in listing a study that has not yet presented results. There are also three studies from China reported, with unspecified time lengths. This seems extraordinary. Have the authors had the papers translated, or have they relied on an abstract? Of these studies, 2 report big effects and one reports almost no effect. If we exclude these studies, there are only 2 that can be compared. These report only modest effects (13% and 20% reduction). The authors need to deal in more detail with the study by Yam et al, which reported on 0.05% atropine. This concentration had a much greater effect (around 67% reduction), and in my opinion seems likely to become the standard dose used. But clearly, more work on the best dose is needed.
#AU1: The authors appreciate your comment. We remove three studies from China. Included paragraph on the study by Yam et al.
#RV2: In relation to OK, there seems to be considerable variation in the efficacy. The paper by Lyu et al seems to report a negative effect. Can the authors check to make sure they have got all values correct? What have meta-analyses of OK results reported.
#AU2: Typo error. Values were exchanged. We made the change in the manuscript.
#RV3: In relation to combinations, the effects seem to be all over the place, and I do not think that the simple conclusion that combinations work is justified.
#AU3: The authors appreciate your comment. The effect of myopia control is present in both atropine and OK, but Table 3 shows that the synergistic effect of both treatments increases the efficacy. Recently, Zhao et al.[1] studied the one-month change in subfoveal choroidal thickness of myopic children treated with 0.01 atropine, OK and their combination and they found that the combination of atropine and OK revealed a greater increased in subfoveal choroidal thickness than monotherapy of atropine alone, which indicate a better myopia control treatment option.
#RV4: The background to this paper is not strong. The authors cited a paper by Zadnik et al on increases in the prevalence of myopia world-wide, but this is a paper on prediction of myopia onset from one set of data. The next paper cited is that of Holden et al., which is more relevant, but it is a set of projections, based on a predictive model that I believe is highly flawed.
#AU4: According to your suggestions. Zadnik et al. was changed to Pan et al.[2] and Holden et al. was removed.
#RV5: I do not understand what the authors are trying to say in lines 31-32, and why it is crucial to make predictions at the age of 8 or 9.
#AU5: Rewritten to improve comprehension.
#RV6: The authors then state that “the slow progress of myopia treatments makes this an essential area for research.” IF the authors mean the limited progress in developing preventive myopia treatments, then I think this gives a false picture of the literature. With OK, 0.05% atropine, MiSight contact lenses and DIMS spectacles (with two other commercial versions about to be released), I think the reverse is true.
#AU6: The authors appreciate your comment and this sentence has been rewritten.
#RV7: The authors then claim that the mechanisms underlying the effects are almost certainly different. I disagree. There are likely to be final common pathways, as is seen for the apparently diverse effects of FDM, LIM and atropine at the scleral level. For optical interventions to affect axial elongation, they need to results in biochemical changes in cellular pathways, and these may be common to all interventions.
#AU7: The authors appreciate your comment and support the argument that there are common pathways at the scleral level. Several articles agree that certain molecular and biochemical events take place in the sclera during the remodeling process, causing structural and biomechanical changes in the tissue and significant thinning of scleral tissue.[3,4]
#RV8: The section on the mechanism of action of atropine is confused. The leading possibility is that atropine produces changes in the IPL, which results in suppression of the increases in EgR1 which are seen when axial elongation increases. Everything flow from there. But it is not clear that the receptors are muscarinic (see reports by Stell and colleagues and Ashby and colleagues).Nickla et al have reported direct effects of atropine on scleral fibroblasts. I think the literature currently regards 0.01% and 0.05% atropine as low doses.
#AU8: The authors appreciate your comment and part of atropine mechanism of action has been rewritten.
#RV9: The authors state that previous studies showed that 0.01% atropine did not significantly decelerate axial elongation, but Table 1 reports on two studies that did show significant differences. Something does not make sense.
#AU9: According to your comment, that sentence contradicts the information in the Table. It has been removed.
#RV10: The claimed difference between Asian and White children is there when absolute changes are considered. But when percentage changes are considered, the effects are the same. This is simply due to the greater progression seen in Asian children, and is not a fundamental difference in drug efficacy.
#AU10: The authors appreciate your comment and this information is considered.
#RV11: In summary, this paper is still confused and incomplete, although it has improved.
#AU11: We took into consideration all reviewers comments and addressed them in the paper. In the same way we hope that the message of the review has been clarified
References
- Zhao, W.; Li, Z.; Hu, Y.; Jiang, J.; Long, W.; Cui, D.; Chen, W.; Yang, X. Short-term effects of atropine combined with orthokeratology (ACO) on choroidal thickness. Contact Lens Anterior Eye 2020, doi:10.1016/j.clae.2020.06.006.
- Pan, C.W.; Ramamurthy, D.; Saw, S.M. Worldwide prevalence and risk factors for myopia. Ophthalmic Physiol. Opt. 2012, doi:10.1111/j.1475-1313.2011.00884.x.
- Upadhyay, A.; Beuerman, R.W. Biological Mechanisms of Atropine Control of Myopia. Eye Contact Lens 2020, 46, 129–135, doi:10.1097/ICL.0000000000000677.
- Metlapally, R.; Wildsoet, C.F. Scleral Mechanisms Underlying Ocular Growth and Myopia. In Proceedings of the Progress in Molecular Biology and Translational Science; 2015.
Reviewer 2 Report
General Comment:
The English is much improved, though (as you will see below) I’m sure that it can be improved still more.
The opinions can be better supported by adding some reference citations, here and there in the text.
I have a few concerns about statements made in the MS (also as you will see below)
Specific Editorial Notes:
[line 14]: “state-of-the-art review” needs to be changed here, as it was changed in the title
[21]: I would prefer, “period; therefore, longer-term …”
[44]: Delete “rate”
[46]: Change “like” to “such as”
[47]: Change “progression” to “progress”
[48]: Change “Within” to “Among”
[51-52]: It would be appropriate here to indicate that the success rates of both orthokeratology and low-dose atropine treatments are less than 100%; therefore, because the mechanisms of these two treatments are almost certainly different, it is possible that the efficacy of these treatments in combination might be greater than that of either one alone.
[57]: Delete 6th word, “the” à “and axial length”
[59]: Change “signal” to “signalling”
[61]: Change “eye system” to “ocular”
[62]: Restore “the”, into “On the one hand” (it pairs with “On the other hand”, in the next sentence)
[65]: Change to “decrease in amount” of collagen
[67]: “the main way to arrest myopia progression is to strengthen the scleral stroma, by increasing its collagen content”: REALLY? I know many myopia scientists who would take issue with this. Perhaps you could say, instead, “one way … might be … “
[63-69]: These lines are heavy on opinion or interpretation; it would be wise to add reference-citations to support them.
[71-73]: This “One theory” – do you agree with it, or not? It’s important, in a review, to make critical judgements that will help the non-expert reader to make important (in this case, clinical) decisions.
[74]: Comma after “topically”
Note: These are the deviations from what I judge to be correct, scholarly English writing, in just the first 2 pages of the manuscript. You deserve a better result for your money.
[242-243], “it has been suggested that children with larger pupil sizes would receive a greater proportion of peripheral myopic defocus associated with OK”: Do you believe this? As per my comment on [71-73], I think that you need to provide critical evaluation here.
[251]: I believe that the word is “coma”, not “comma”
[256]: “authors like”: No, Guo et al. are the only ones that you cite here.
[272}: Change “others” to “other”
Again: THESE ARE NOT THE ONLY ERRORS IN THE EDITING FOR ENGLISH, BUT MERELY EXAMPLES.
[338], “efficacy has a synergic [I prefer “synergistic”] effect”: This means that they are more than simply additive; what is the evidence for that?
[341], “in other populations than just Asian populations”: You can simplify to, “in other than Asian populations”
[342]: I think that you might mean “evidence”, rather than “research”
Author Response
#RV1: The English is much improved, though (as you will see below) I’m sure that it can be improved still more. The opinions can be better supported by adding some reference citations, here and there in the text. I have a few concerns about statements made in the MS (also as you will see below)
#AU1: We would like to thank reviewers for their insightful comments on the paper, as these comments led us to improve it. We took into consideration all reviewers comments and addressed them in the paper. Detailed responses to reviewers are given below.
#RV2: [line 14]: “state-of-the-art review” needs to be changed here, as it was changed in the title
#AU2: Correction made
#RV3: [21]: I would prefer, “period; therefore, longer-term …”
#AU3: Correction made
#RV4: [44]: Delete “rate”
#AU4: Correction made
#RV5: [46]: Change “like” to “such as”
#AU5: Correction made
#RV6: [47]: Change “progression” to “progress”
#AU6: [48]: Change “Within” to “Among”
#RV7: [51-52]: It would be appropriate here to indicate that the success rates of both orthokeratology and low-dose atropine treatments are less than 100%; therefore, because the mechanisms of these two treatments are almost certainly different, it is possible that the efficacy of these treatments in combination might be greater than that of either one alone.
#AU7: appreciation included
#RV8: [57]: Delete 6th word, “the” à “and axial length”
#AU8: Correction made
#RV9: [59]: Change “signal” to “signalling”
#AU9: Correction made
#RV10: [61]: Change “eye system” to “ocular”
#AU10: Correction made
#RV11: [62]: Restore “the”, into “On the one hand” (it pairs with “On the other hand”, in the next sentence)
#AU11: Correction made
#RV12: [65]: Change to “decrease in amount” of collagen
#AU12: Correction made
#RV13: [67]: “the main way to arrest myopia progression is to strengthen the scleral stroma, by increasing its collagen content”: REALLY? I know many myopia scientists who would take issue with this. Perhaps you could say, instead, “one way … might be … “
#AU13: Correction made
#RV14: [63-69]: These lines are heavy on opinion or interpretation; it would be wise to add reference-citations to support them.
#AU14: Zou et al.[1] reference was added.
#RV15: [71-73]: This “One theory” – do you agree with it, or not? It’s important, in a review, to make critical judgements that will help the non-expert reader to make important (in this case, clinical) decisions.
#AU15: The authors agree with this theory, correction made.
#RV16: [74]: Comma after “topically” Note: These are the deviations from what I judge to be correct, scholarly English writing, in just the first 2 pages of the manuscript. You deserve a better result for your money.
#AU16: Correction made
#RV17: [242-243], “it has been suggested that children with larger pupil sizes would receive a greater proportion of peripheral myopic defocus associated with OK”: Do you believe this? As per my comment on [71-73], I think that you need to provide critical evaluation here.
#AU17: Correction made
#RV18: [251]: I believe that the word is “coma”, not “comma”
#AU18: Correction made
#RV19: [256]: “authors like”: No, Guo et al. are the only ones that you cite here.
#AU19: Correction made
#RV20: [272}: Change “others” to “other”. Again: THESE ARE NOT THE ONLY ERRORS IN THE EDITING FOR ENGLISH, BUT MERELY EXAMPLES.
#AU20: Correction made
#RV21: [338], “efficacy has a synergic [I prefer “synergistic”] effect”: This means that they are more than simply additive; what is the evidence for that?
#AU21: Correction made, evidence was presented on Table 3.
#RV22: [341], “in other populations than just Asian populations”: You can simplify to, “in other than Asian populations”
#AU22: Correction made
#RV23: [342]: I think that you might mean “evidence”, rather than “research”
#AU23: Correction made
- Zou, L.; Liu, R.; Zhang, X.; Chu, R.; Dai, J.; Zhou, H.; Liu, H. Upregulation of regulator of G-protein signaling 2 in the sclera of a form deprivation myopic animal model. Mol. Vis. 2014.